# Spatiotemporal Variability of Surface Phytoplankton Carbon and Carbon-to-Chlorophyll a Ratio in the South China Sea Based on Satellite Data

Wenlong Xu [1,2], Guifen Wang [1,2,*], Long Jiang [1,2], Xuhua Cheng [1,2], Wen Zhou [3] and Wenxi Cao [3]

[1] Key Laboratory of Marine Hazards Forecasting, Ministry of Natural Resources, Hohai University, Nanjing 210098, China; wenlongxu@hhu.edu.cn (W.X.); ljiang@hhu.edu.cn (L.J.); xuhuacheng@hhu.edu.cn (X.C.)
[2] College of Oceanography, Hohai University, Nanjing 210098, China
[3] State Key Laboratory of Tropical Oceanography, South China Sea Institute of Oceanology, Chinese Academy of Sciences, Guangzhou 510301, China; wenzhou@scsio.ac.cn (W.Z.); wxcao@scsio.ac.cn (W.C.)
* Correspondence: guifenwang@hhu.edu.cn

**Abstract:** The spatiotemporal variability of phytoplankton biomass has been widely studied because of its importance in biogeochemical cycles. Chlorophyll a (Chl-a)—an essential pigment present in photoautotrophic organisms—is widely used as an indicator for oceanic phytoplankton biomass because it could be easily measured with calibrated optical sensors. However, the intracellular Chl-a content varies with light, nutrient levels, and temperature and could misrepresent phytoplankton biomass. In this study, we estimated the concentration of phytoplankton carbon—a more suitable indicator for phytoplankton biomass—using a regionally adjusted bio-optical algorithm with satellite data in the South China Sea (SCS). Phytoplankton carbon and the carbon-to-Chl-a ratio ($\theta$) exhibited considerable variability spatially and seasonally. Generally, phytoplankton carbon in the northern SCS was higher than that in the western and central parts. The regional monthly mean phytoplankton carbon in the northern SCS showed a prominent peak during December and January. A similar pattern was shown in the central part of SCS, but its peak was weaker. Besides the winter peak, the western part of SCS had a secondary maximum of phytoplankton carbon during summer. $\theta$ exhibited significant seasonal variability in the northern SCS, but a relatively weak seasonal change in the western and central parts. $\theta$ had a peak in September and a trough in January in the northern and central parts of SCS, whereas in the western SCS the minimum and maximum $\theta$ was found in August and during October–April of the following year, respectively. Overall, $\theta$ ranged from 26.06 to 123.99 in the SCS, which implies that the carbon content could vary up to four times given a specific Chl-a value. The variations in $\theta$ were found to be related to changing phytoplankton community composition, as well as dynamic phytoplankton physiological activities in response to environmental influences; which also exhibit much spatial differences in the SCS. Our results imply that the spatiotemporal variability of $\theta$ should be considered, rather than simply used a single value when converting Chl-a to phytoplankton carbon biomass in the SCS, especially, when verifying the simulation results of biogeochemical models.

**Keywords:** chlorophyll a; phytoplankton carbon; carbon-to-chlorophyll a ratio; South China Sea; satellite data

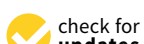



## 1. Introduction

Phytoplankton form the base of marine ecosystems. More than half of the primary production on Earth occurs in the surface layer of the global ocean and involves photosynthetic fixation of carbon by phytoplankton [1,2]. These primary producers are also fundamental players in marine biogeochemical cycles [3,4]. Phytoplankton biomass and productivity show variations in timing and magnitude in different regions from processes affecting their growth and demise—such as nutrient supply, incident solar irradiance, and

predation pressure—that may vary with latitude, apart from regional and local oceanographic conditions. Due to these important and complicated processes, it is important to study the distribution of phytoplankton biomass with high spatial and temporal resolution.

The most widely used predictor for monitoring phytoplankton biomass is chlorophyll a concentration (Chl-a), and it is still the most common measure of phytoplankton in ecological studies. Chl-a can be estimated remotely using ocean color satellites [5], in situ with fluorometers [6] or radiometers, or measured on discrete samples through high performance liquid chromatography [7], fluorometric [8] or spectrophotometric methods [9]. Even though Chl-a is widely used as a proxy, the relationship between Chl-a and phytoplankton biomass is not straightforward. It has long been recognized that variability in intracellular Chl-a content from light acclimation, nutrient stress, phytoplankton community structure, and other environmental stresses confounds the relationship between chlorophyll and phytoplankton biomass [10–14].

Phytoplankton carbon—as an alternative index—might provide a more accurate distribution of phytoplankton biomass and improve our ability to estimate primary productivity. Much effort had been done to develop approaches to retrieve phytoplankton carbon from optical measurements [15], including satellite ocean color observations [16]. Particulate backscattering coefficients ($b_{bp}$, m$^{-1}$) in marine waters are shown to covary with suspended particles [17] and can be obtained from satellite remote sensing [18]. $b_{bp}$ have been used to estimate phytoplankton carbon in the open ocean (Behrenfeld et al. 2005; phytoplankton carbon = $13{,}000 \times (b_{bp} - 0.00035)$, where the background value is 0.00035 m$^{-1}$ and scale factor is 13000 mg m$^{-2}$). The contribution of nonalgal particles to the backscattering coefficient ($b_{bpNAP}$) is a constant value of 0.00035 m$^{-1}$ in this estimate [16]. The underlying assumption is that $b_{bpNAP}$ represents the stable heterotrophic and detrital components of the surface population and is therefore not dependent on phytoplankton dynamics. However, Siokou-Frangou et al. (2010) [19] showed that the heterotrophic and detrital components of the surface particle population in the Mediterranean Sea are neither negligible [20] nor stable, but highly dynamic in both space and time. In addition, Bellacicco et al. (2016, 2018, 2019, 2020) [21–24] found that $b_{bpNAP}$ were highly variable using satellite data and Bio-Argo data in the Mediterranean Sea and globally. Therefore, in order to calculate phytoplankton carbon concentrations accurately, the regionally adjusted $b_{bpNAP}$ should be considered when using the phytoplankton carbon retrieval algorithm.

The carbon-to-Chl-a ratio of phytoplankton (hereafter termed $\theta$ following Geider et al. 1987 [11]) plays an essential role in the computation of primary production [25], in evaluating the phytoplankton community dynamics and its physiological adjustments to environmental factors [13,26], and in verification and initialization of global biogeochemical models [27,28]. Moreover, the carbon-to-Chl-a ratio of phytoplankton is a dynamic and highly variable property [29]. Estimating such changes in $\theta$ in response to variations in available light, nutrient, and temperature, is not a trivial task, but it is an essential step in primary production models and biogeochemical models. A wide range of $\theta$ values has been reported in the literature across laboratory, field, and model studies [11,25,29–32]. Sathyendranath (2009) [30] provided a conservative estimate of $\theta$ (15–176) based on field data collected across different types of marine and coastal environments, and proposed an empirical model to estimate phytoplankton carbon from Chl-a. Based on biogeochemical model results, Wang (2009) [31] observed $\theta$ within the mixed layer increased from <100 in the eastern upwelling region to >150 in the warm pool in the Pacific, whereas subsurface $\theta$ was about 50 below 100 m water depth. Seasonal variation of $\theta$ has also been documented from a large amount of field measurements [32] which showed that the lowest $\theta$ during winter ($\approx$15) and summer $\theta$ between 20 and 96. With the advance of ocean color remote sensing, the large-scale and long-term variability of $\theta$ could be estimated from satellite data, which were expected to significantly enhance our understanding of $\theta$.

The South China Sea (SCS) is the largest tropical marginal sea in the northwestern Pacific Ocean. It is a semiclosed deep sea with an average water depth over 2000 m. The East Asian monsoon is characterized by two regimes, the southwest monsoon from late

May to September and the northeast monsoon from November to March, which plays an important role in the hydrological features and upper circulation of the SCS [33–36]. The SCS basin is quite similar to oligotrophic regions with the primary producers dominated by picophytoplankton [37–39]. The temporal and spatial distribution of physical, ecological, and biogeochemical parameters in the SCS and their driving mechanisms have been research foci for many years. For example, Ning et al. (2004) [37] used multidisciplinary survey data to describe the phytoplankton seasonal distribution and revealed clearly coupled physical–chemical–biological oceanographic processes in relation to phytoplankton abundance and production in the SCS. Additional studies used satellite derived Chl-a data to investigate the spatial and temporal distributions of phytoplankton in the SCS and found significant seasonal change, with high concentration in winter and low values in summer, and higher Chl-a in inshore areas than that in the central sea basin [39–45]. In the northern SCS, Chl-a starts to increase in September and reaches its maximum in December or January [46]. Furthermore, seasonal and spatial variability of primary production and phytoplankton carbon has been studied in the SCS using physical–biogeochemical model data [47,48].

However, study on the phytoplankton biomass in units of carbon are quite limited with Chl-a usually used as the only indicator of phytoplankton biomass in the SCS. We know less about variability of carbon-to-chlorophyll ratio which was affected by many environmental stressors such as light and nutrient limitation. Therefore, the objective of this study was to estimate phytoplankton carbon from the backscatter-based model using satellite, and then to investigate the spatial and seasonal variability of phytoplankton carbon and its ratio to Chl-a by accounting for physiological environment factors and phytoplankton community structures.

## 2. Materials and Methods

### 2.1. Study Area and Sampling

Multiyear Chl-a concentrations were measured at 133 stations from different regions in the SCS (Figure 1). The regions included coastal high productivity waters and the oligotrophic open ocean. The range of Chl-a concentrations was about 0.03–2.32 mg m$^{-3}$.

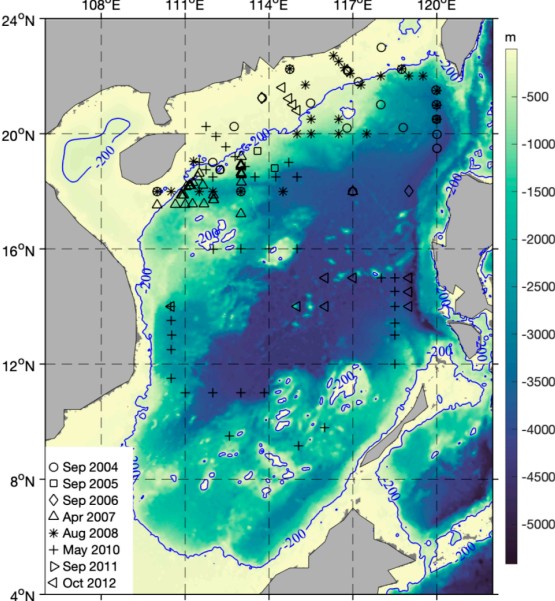

**Figure 1.** Hydrographic stations at which chlorophyll a (Chl-a) concentrations were measured in the South China Sea from 2004 to 2012. The survey stations from different cruises are distinguished by different marker types. The blue line shows the 200-m isobath.

Seawater samples were collected in clean Niskin bottles on a rosette with a conductivity–temperature–density instrument. A volume of water (0.5–1 L) was filtered onto glass fiber filters (Whatman GF/F) at low vacuum. Filters were stored frozen at −80 °C before analysis in the laboratory. The Chl-a concentrations were measured with a Turner-Design 10-AU Fluorometer.

### 2.2. Satellite Data

We used a continuous time-series of global ocean color data—produced by the European Space Agency's Ocean Colour Climate Change Initiative (OC-CCI) project (https://www.oceancolour.org/)—by systematically merging available satellite data from three major sensors: NASA-SeaWiFS, NASA-MODIS-Aqua, and ESA-MERIS [49]. In version 4.2, the best algorithms for Chl-a retrieval were selected depending on water class memberships and the particulate optical backscattering coefficient at 555 nm were produced by the Quasi-Analytical algorithm (QAA) [18]. The diffuse attenuation coefficient at 490 nm ($K_{d490}$) and the photosynthetically active radiation (PAR) were downloaded from GlobColour (http://hermes.acri.fr/). All of the variables reported monthly values from September 1997 to December 2019 at a spatial resolution of 4 × 4 km. The sources and types of particulate matter that affect Chl-a and $b_{bp}$ are more complicated in the near-shore shallow water of the SCS (<200 m water depth). Therefore, we focused on the open ocean with water depths deeper than 200 m.

The sea surface temperature (SST) data are from the Optimum Interpolation Sea Surface Temperature (OISST) dataset provided by the NOAA Earth System Research Laboratory (https://www.ncdc.noaa.gov/). Monthly OISST data (data from the AVHRR sensor only with the longest record) from September 1997 to December 2019 were used in this study.

Over the photoperiod, incident light in the mixed layer decreases exponentially with water depth. To reduce its impact, values of mean light in the mixed layer ($I_g$) were calculated by:

$$I_g = PARe^{-0.5K_{d490}MLD} \tag{1}$$

where *MLD* is mixed layer depth. The monthly mean climatology of *MLD* data were downloaded from the SCS Physical Oceanographic Dataset (SCSPOD14), which is based on 51,392 validated temperature and salinity profiles for the period 1919–2014 [50]. *MLD* is defined as the depth where the potential density is 0.03 kg m$^{-3}$ greater than the reference depths as 10 m.

### 2.3. Estimates of Phytoplankton Carbon

Phytoplankton carbon was retrieved using the formula proposed by Behrenfeld et al. (2005) [16]:

$$\text{Carbon(x, y, t)} = \left[ b_{bp}(x, y, t) - b_{bpNAP} \right] SF \tag{2}$$

where *x* and *y* are longitude and latitude, *t* is the time in months, the conversion coefficient (*SF*) was 13,000 mg C m$^{-2}$ and $b_{bpNAP}$ is the part of $b_{bp}$ contributed by nonalgal particles. It should be pointed out that when retrieving $b_{bpNAP}$, Behrenfeld et al. (2005) used $b_{bp}$ at 440 nm, but $b_{bp}$ at 555 nm was used in this study. Therefore, a spectral dependence of $\lambda^{-1.03}$ was applied to transfer the curve reported at 440–555 nm consistent with the slope used by GSM model [51]. The $b_{bpNAP}$ was estimated as the intercept of the least square linear regression fit between Chl-a and $b_{bp}$. A new value of $b_{bpNAP}$ (0.00056 m$^{-1}$) was estimated by using the monthly mean climatology of Chl-a and $b_{bp}$(555) in the SCS (Figure 2). Our value is relatively higher than the global mean value estimated by Behrenfeld et al. (2005). As a result, the phytoplankton carbon concentrations retrieved by Eq. 2 and with regionally adjusted $b_{bpNAP}$ values are only one-half of previously published values. This difference has also been documented in a previous study [22].

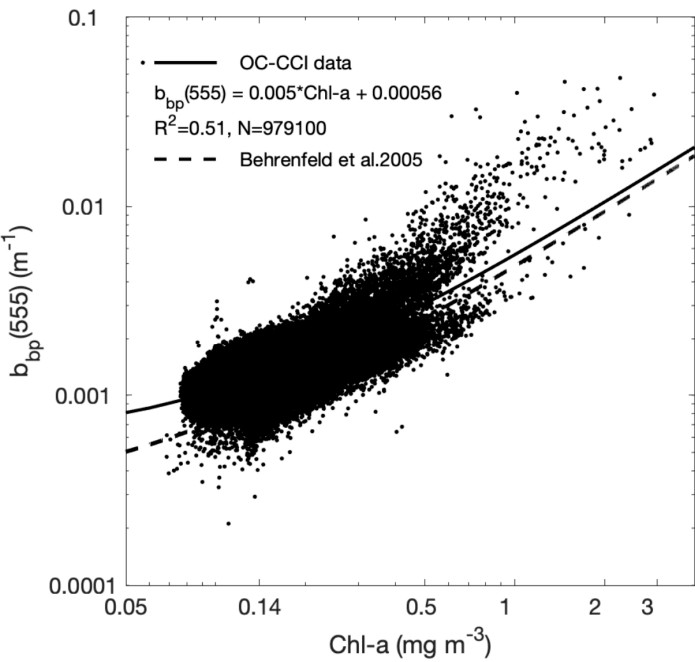

**Figure 2.** Particle backscattering at 555 nm ($b_{bp}$(555), m$^{-1}$) is likely a function of chlorophyll a concentration (Chl-a, mg m$^{-3}$) for monthly mean climatology of satellite data (September 1997 to December 2019) of the South China Sea. The solid black line represents a linear fit to (Ocean Colour Climate Change Initiative) OC-CCI data. For comparisons, a similar relationship obtained from global satellite dataset [16] (black dash line) is shown. We applied a spectral dependence of $\lambda^{-1.03}$ to transfer the curve reported at 440–555 nm consistent with the slope used by the GSM model [51].

### 2.4. Algorithm to Estimate Phytoplankton Size Classes

Based on 7 years of pigment measurements acquired in the SCS, Lin et al. (2014) re-parameterized a three-component model of phytoplankton size classes (PSC) [52]. The model could be expressed as:

$$F_m = \frac{C - C_{p,n}^m \times \left[1 - exp(-S_{p,n}C)\right]}{C} \tag{3}$$

$$F_n = \frac{C_{p,n}^m \times \left[1 - \exp(-S_{p,n}C)\right] - C_p^m \times \left[1 - \exp(-S_pC)\right]}{C} \tag{4}$$

$$F_m = \frac{C_p^m \times \left[1 - \exp(-S_pC)\right]}{C} \tag{5}$$

where, $C$ was the total Chl-a concentration, and $C_{p,n}^m$ (mg m$^{-3}$), $C_p^m$ (mg m$^{-3}$), $S_{p,n}$, and $S_p$ were statistical parameters derived from fitting model to in situ data with values suitable for SCS that are 0.9532, 0.2563, 0.9835, and 3.5346, respectively [53]. The model was relatively accurate for the inversion of PSC in the SCS, with the relative root mean squared errors being 28.2% (pico-plankton), 20.8% (nano-plankton), and 18.8% (micro-plankton), respectively.

### 2.5. Dividing Biogeochemical Provinces in the SCS

In previous studies, the time series of remote sensing observations were usually calculated by averaging the pixels data in a given part of the study area [40,44,54]. This part is usually defined in terms of geography or pure geometry by assuming that study area is highly uniform, and its main characteristics change slightly, which may be less instructive for areas with large variability. Instead, in this study, trophic regimes of the South China Sea were classified based on the seasonality cycle of surface Chl-a data from satellite. As some researchers have reported in Mediterranean Sea and Indian Ocean [55,56], seasonality of

surface Chl-a could reflect different mechanisms driving the functioning of the ecosystem, like, the increase of Chl-a appears to be strongly coupled to physical forcing which could induce locally favorable conditions for phytoplankton growth.

Monthly climatological time-series of Chl-a were then engendered for each pixel of the SCS. The resulting climatological time-series were then normalized by the maximum values of each specific pixel. Then, k-means clustering was implemented for the normalized data with the aim to statistically organize the time-series dataset and to generate clusters representing regions of similarity. One of the key challenges for the k-means algorithm was determining the number of clusters in a data set. Here, the number of clusters was determined by four specific tests [57] conducted on satellite image data. The optimal number of clusters was six, which was determined through use of a series of evaluation indicators in our dataset.

In general, the structure and spatial classification of the SCS pixels—as obtained with the k-means method—were very consistent (Figure 3a). The geographical boundaries between clusters are well defined and the heterogeneity within clusters is very low. The temporal evolution of the six cluster centers can be seen in Figure 3b. Noticeably, the constraints imposed on to the k-means algorithm were (1) the number of clusters and (2) the normalization implemented on the climatological time-series.

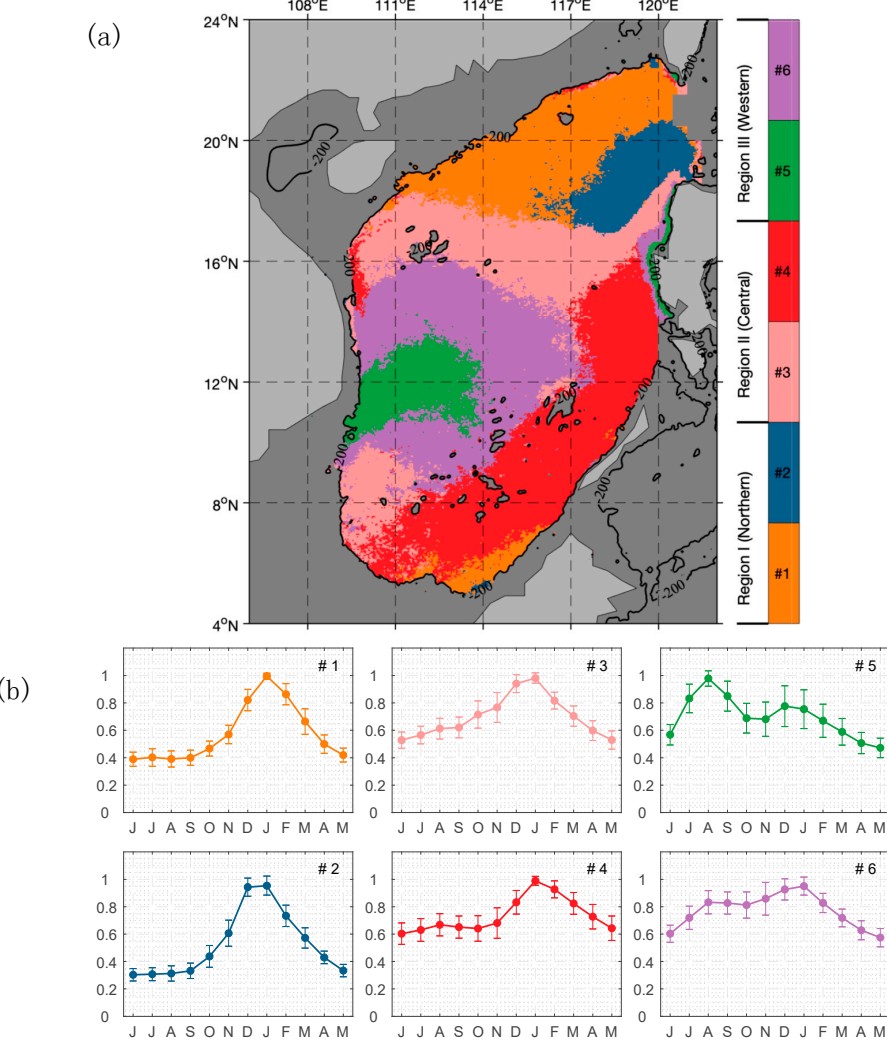

**Figure 3.** Results of k-means cluster analysis showing (**a**) spatial distribution of the clusters (1–6) and (**b**) temporal evolution of the cluster centers. The black lines in (**a**) show the 200-m isobath. The curve colors in (**b**) correspond to the same color scale in (**a**). Error bars in (**b**) indicate the relative +/− one standard deviation.

To better describe the results, clusters were then regrouped into three regions using characteristics of the cluster center temporal evolution. Specifically, two clusters (#1 and #2, Region I, Figure 3)—located in the northern of the SCS—almost covers the area north of 17° N (about 37% of the total area). Its annual pattern indicates a maximum in January with appreciable growth in September and loss in February. The slight difference within the two clusters is that the magnitude of Cluster #1 was about 1.5 times greater than Cluster #2. Another two clusters (#3 and #4, Region II, Figure 3) are widely distributed in the central part of SCS including the deep basin on the Philippine side of the SCS and the SCS central basin along 15°N. The most obvious characteristic of those clusters is a rapid increase in December, which then peaked in January. However, the seasonal signals in these categories are very weak with very similar values all year round. Additionally, the southeastern area offshore Vietnam—the offshore extension of the Vietnam coastal current—were categorized as one region (#5 and #6, Region III, Figure 3), denoted as the western part of SCS. This cluster experiences two peaks of Chl-a throughout the year—one in August and the other in January.

*2.6. Statistical Methods*

To quantitatively assess the difference between the OC-CCI derived Chl-a data and in situ Chl-a data, several statistical parameters were used: (1) the bias (the average difference), (2) the root mean squared error, and (3) the median absolute relative difference.

The bias is defined as:

$$bias = \frac{1}{N} \sum_{i=1}^{N} (Chl_{sat} - Chl_{insitu}) \tag{6}$$

the root mean squared error (RMSE) as:

$$RMSE = \sqrt[2]{\frac{1}{N} \sum_{i=1}^{N} (Chl_{sat} - Chl_{insitu})^2} \tag{7}$$

and the median absolute relative difference (MARD) is expressed in percentage as:

$$MARD = 100 \times median \left[ \frac{|Chl_{sat} - Chl_{insitu}|}{Chl_{insitu}} \right] \tag{8}$$

where $Chl_{sat}$ and $Chl_{insitu}$ are satellite derived and in situ Chl-a values, respectively; $N$ is the number of matched up data points.

Statistical analysis was used to assess the existence of linear dependence between $\theta$ and other environmental variables. The correlation and its significance were calculated using linear regression and Student's *t*-tests following Santer et al. (2000) [58].

# 3. Results

*3.1. Validation of OC-CCI Product*

The correlation between Chl-a concentrations measured shipboard and those estimated from satellite data for each station is shown in Figure 4. To minimize the differences between in situ measured Ch-a concentrations and those estimated from satellite data, it is important to use satellite data for the pixel closest to the station and obtained as soon as possible after sampling at the station. The mean values within the $3 \times 3$ pixel area of daily satellite data were selected to match in situ measurement. Statistical comparisons show relatively good agreement between in situ data and the satellite data, with a statistically significant correlation ($R^2 = 0.71$, $p < 0.001$), a low median absolute relative difference (MARD = 36.99%), and a low root mean square error (RMSE = 0.23 mg m$^{-3}$). OC-CCI data are slightly higher than in situ Chl-a with the bias being around 0.04 mg m$^{-3}$ (Figure 4). The validation results present here provide confidence in the use of OC-CCI Chl-a data in the SCS.

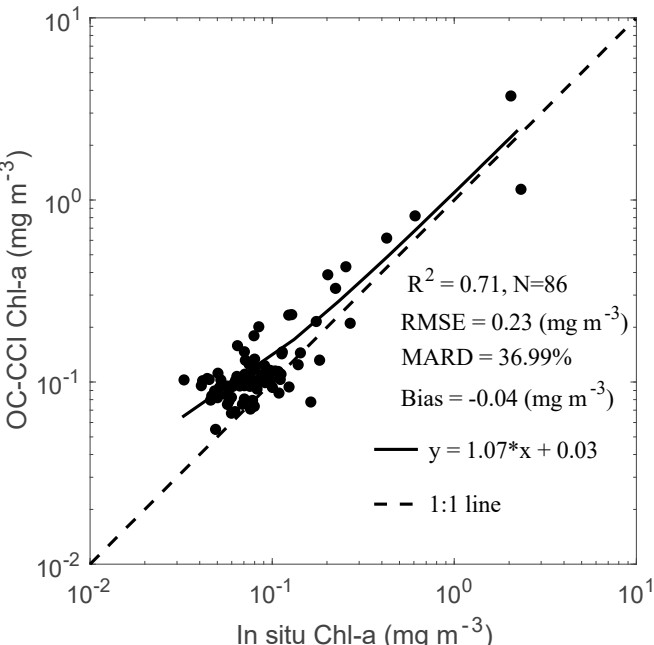

**Figure 4.** Correlation between Chl-a concentrations measured shipboard and those estimated from OC-CCI data for each hydrographic station. Selected criterium OC-CCI data were used, which were averaged within a 3 × 3 pixel box of daily satellite collocation. Owing to persistent cloud cover, the daily products resulted in no data being available for 47 of the 133 stations.

The use of $b_{bp}$ values retrieved by QAA has been widely used in ocean ecology and biogeochemistry research [16,59,60] and has been validated globally and on a regional scale [61,62]. In the SCS, the $b_{bp}$ was validated by a limited set of in situ bio-optical data that showed the inversion values were close to the measured value, as indicated by a low mean absolute percent error difference (32.08%) and a high coefficient of determination (0.92) [62].

### 3.2. Comparison of Phytoplankton Carbon Estimates Using Different Empirical Algorithms

To evaluate the phytoplankton carbon determined by the empirical algorithm, we plotted the satellite-retrieved phytoplankton carbon as a function of Chl-a and added the existing empirical relationships between Chl-a and phytoplankton carbon (Figure 5). The solid black line representing the best fit was plotted in Figure 5, and it was consistent with the model proposed by Maranon et al. (2014) [63] and using in situ data collected in surfaced waters over coastal and open-ocean regions, and consistent with the model developed by Loisel et al. (2018) [64] based on satellite data. However, the relationships proposed by Buck et al. (1996) [65] for satellite-derived phytoplankton carbon as a function of observational Chl-a and Sathyendranath et al. (2009) [30] for a simple conceptual model to infer in situ phytoplankton carbon as a function of Chl-a differ significantly from our results (Figure 5). Their values are higher when Chl-a concentrations were <1 mg m$^{-3}$ and are lower when Chl-a were >1 mg m$^{-3}$. After comparing our results with published empirical relationships, they suggest that phytoplankton carbon retrieved by our adjusted algorithm is more reliable.

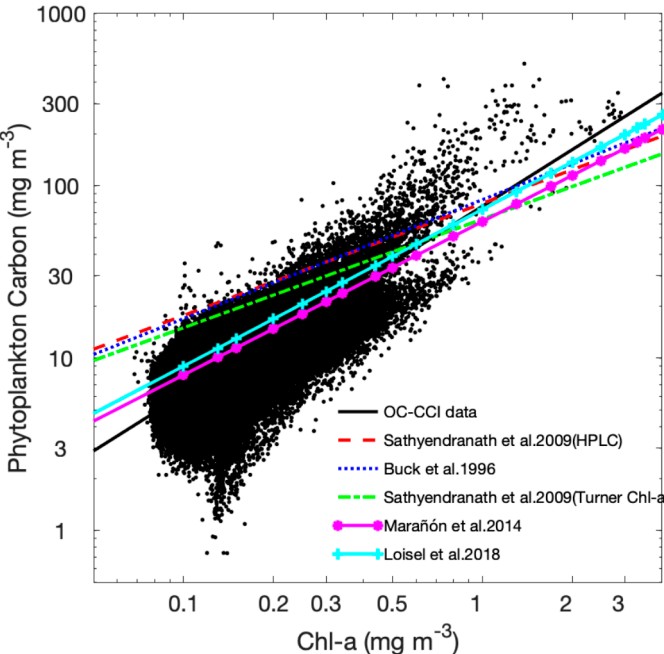

**Figure 5.** Phytoplankton carbon retrieved with a modified method proposed by Behrenfeld et al. (2005) as a function of Chl-a for monthly mean climatology of satellite data (September 1997 to December 2019) of the South China Sea. The solid black line represents a power law fit to OC-CCI data. For comparison, we also included several empirical fits. The red dashed and blue dash-dotted curves correspond to Sathyendranath et al. (2009) for HPLC and Turner Chl-a, respectively. The other empirical algorithms derived by Buck et al. (1996) (blue dotted line), Mara et al. (2014) (magenta solid line with asterisk symbol), and Loisel et al. (2018) (cyan solid line with plus sign symbol) are also shown.

### 3.3. Spatiotemporal Distribution of Phytoplankton Carbon and θ

A 22-year analysis of composite monthly mean phytoplankton carbon showed pronounced regional and seasonal differences in the SCS (Figures 6 and 7). The phytoplankton carbon maps revealed similarities with Chl-a concentrations (Figure A1), which is unsurprising because of the correlation between those properties. Generally, phytoplankton carbon in the northern SCS was higher than in the southern and central areas (Figure 7a–c). The seasonal pattern of phytoplankton carbon showed that maximum values in the northern and central SCS occurred in winter and minimum values in the whole SCS occurred in spring (Figure 6). Interestingly, phytoplankton carbon showed relatively weak seasonality in the deep basin on the Philippine side of the SCS (Figures 6 and 7b), but considerably strong seasonal patterns in the northern SCS (Figure 7a). Individual pixels of phytoplankton carbon of the averaged 22-year period were $9.17\pm6.72$ mg m$^{-3}$. In the summer and early autumn (June to September), phytoplankton carbon concentrations were low throughout the SCS with the exception of higher concentrations to the east of southern Vietnam (Figure 7c).

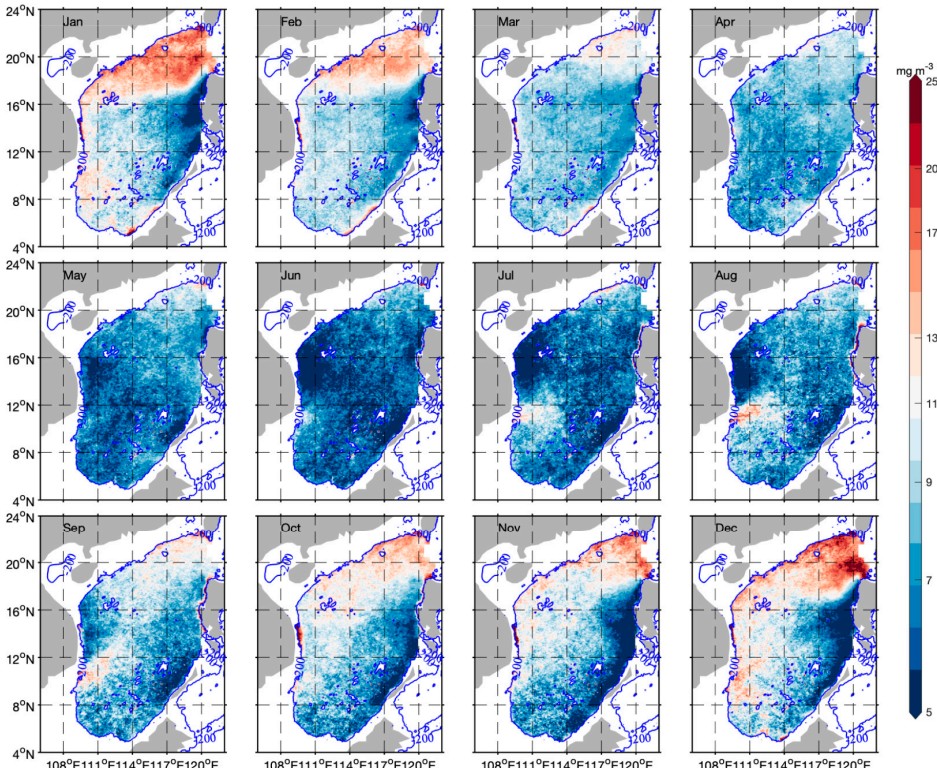

**Figure 6.** Monthly mean climatological distribution of the phytoplankton carbon in the South China Sea (September 1997 to December 2019). The blue line shows the 200-m isobath.

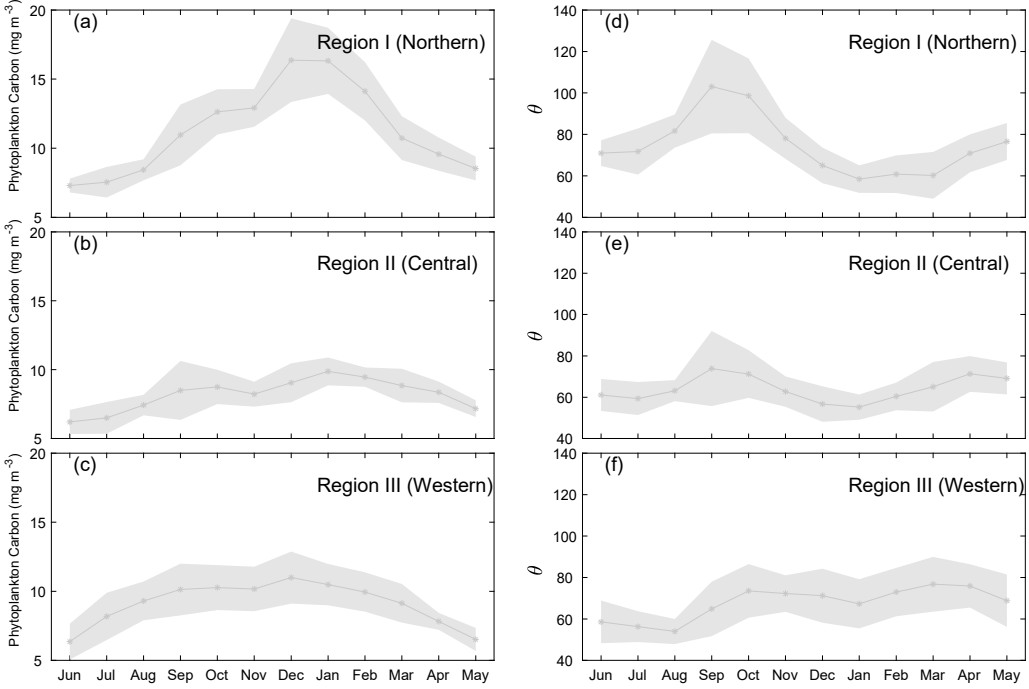

**Figure 7.** Time-series of monthly mean climatology of phytoplankton carbon (**a**–**c**) and $\theta$ (**d**–**f**) in the northern (Region I), central (Region II), and western (Region III) parts of South China Sea. The definitions of those regions are shown in Figure 3a. Color shading indicates standard deviations of the monthly mean climatology of $\theta$. Noting that the time starts in June.

The monthly averaged climatology of $\theta$—computed over the SCS using composite monthly images from September 1997 to December 2019—varied over a wide range from

<20 in eutrophic waters to >90 in oligotrophic regions (Figure 8). These results are consistent with our current understanding of oceanic systems where low and high $\theta$ values represent regions characterized by microplankton and picoplankton, respectively [30,66,67]. The $\theta$ of individual pixels averaged over 22 years was 67.77 ± 32.44. In addition, the monthly mean climatological distribution of $\theta$ in the SCS showed pronounced regional and seasonal variations (Figures 7d–f and 8). The mean $\theta$ in the northern (Region I), central (Region II), and western (Region III) parts of SCS was 64.13 ± 11.08, 74.83 ± 17.98, and 67.73 ± 13.23, respectively. The monthly average climatology of $\theta$ showed highest values during September in the western part of SCS (Region I: 103.05 ± 22.55) and central part of SCS (Region II: 73.8 ± 18.17), but occurred in March in western part of SCS (Region III: 76.74 ± 13.21). The lowest values were observed during January in the northern SCS (Region I: 58.41 ± 6.66) and central SCS (Region II: 55.21 ± 6.12), but in August in the western SCS (Region III: 53.93 ± 0.05). In contrast to the seasonal pattern of phytoplankton carbon, relatively lower $\theta$ values were observed during winter compared to autumn.

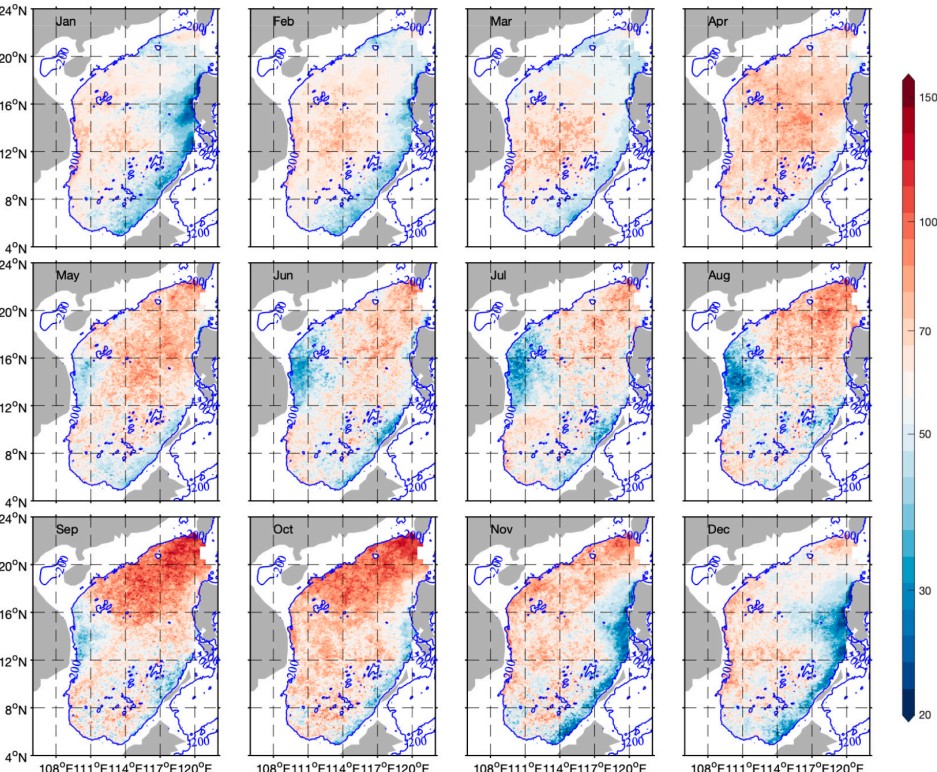

**Figure 8.** Monthly mean climatological distribution of the $\theta$ in the South China Sea (September 1997 to December 2019). The blue line shows the 200-m isobath.

Although there are some regional differences, a seasonal cycle is evident. There is a distinct annual cycle in the western SCS (Region I) where high values last from September to January (Figure 7d). In central SCS (Region II), there are two peaks, one in September and one in April (Figure 7e). It is interesting to note that the summer in the western SCS (Region III) has considerably lower values, while values in other months are higher (Figure 7f).

## 4. Discussion

### 4.1. Spatial and Temporal Variability of Phytoplankton Carbon and $\theta$

Although many studies have been conducted on the spatial and temporal distribution of Chl-a [44,45,68] and particulate organic carbon [69–71] in the SCS using ocean color data, little research has focused on phytoplankton carbon. It has long been recognized that the high concentration of Chl-a in winter in the northern SCS was due to the strong mixing caused by the northeast monsoon [46,72–74]. In our research, the similar seasonal

pattern in the northern SCS was also shown in phytoplankton carbon, with a prominent peak during December–January of the following year. The winter peak of phytoplankton carbon in the central SCS was much weaker than the peak of Chl-a. In the western part of SCS, Chl-a showed a bimodal structure, with one peak in winter and the other in summer and phytoplankton carbon also showed the same distribution characteristics.

The spatial variability and magnitude of $\theta$ retrieved in our study were shown to be relatively consistent with seasonal patterns reported by Xiu and Chai (2012) for the SCS [48]. Xiu and Chai (2012) includes—to our knowledge—the rarely published data describing the annual cycle of $\theta$ for the SCS used biogeochemical model. A large variation wide range of $\theta$ was found in their model results, and the minimum $\theta$ always occurred in winter and high values present in later summer to early autumn. Based on our estimation from ocean color data, a wide range of $\theta$ ($\approx$26–$\approx$130) were found in SCS and similar results about the seasonal distribution of $\theta$ were found in the northern (Region I) and central part (Region II) of SCS. However, in the western SCS (Region III), different seasonal pattern was observed with its lowest value being in summer, which might be influenced by upwelling driven by southwest monsoon [75]. This bias was likely related to their biogeochemical model resolution which cannot fully resolve coastal upwelling condition in the western SCS. Upwelling brought cold and nutrient-repleted water from the depth of the ocean to the nutrient-depleted surface water, which stimulated the growth of phytoplankton and might have potentially altered the phytoplankton community structure (the fraction of microplankton might be increased) [76]. Due to new nutrients added, microphytoplankton, such as diatoms, had more sensitivity response to nutrient uptake than other phytoplankton groups [76]. In addition, a northeastward jet in the western SCS interaction with mesoscale eddy may increase the microphytoplankton contribution through water mass transport and mixing [77]. Therefore, lower values of $\theta$ in the western SCS in the summer may be attributed to the increase of microphytoplankton caused by upwelling. In addition, low temperature caused by upwelling could also influence the $\theta$ value, but its effect was more complex [12,32,78]. High values of $\theta$ in the open ocean—often well above 100—have also been documented by a field study [32] and satellite results [16]. One possible explanation is that in open ocean environments or at low latitudes, a more permanent nutrient limitation leads to relatively high values of $\theta$ [32].

### 4.2. Factors Affecting $\theta$

Intracellular Chl-a varies as a function of many factors, including cell size, growth, ambient light, temperature, and nutrient status. Hence, it is not surprising that $\theta$ varies throughout the world's aquatic systems by three or more orders of magnitude of change [32,79]. The reasons for $\theta$ variability have been explained in this study by considering the mean light intensity within the *MLD* ($I_g$), sea surface temperature (SST), and the phytoplankton community structure.

The phytoplankton $\theta$ may be affected by physiological processes [11,80], which have a great impact on cells because they change intracellular Chl-a concentrations in response to light and nutrient availability. The two main factors that affect the physiological state of phytoplankton—light and nutrient stress—are represented by the mean light in the mixed layer ($I_g$) and SST, respectively [16]. To interpret the variability in photoacclimation and its causes, $\theta$ as a function of SST and $I_g$ are shown in Figures 9 and 10. Quite specifically, increases in $\theta$ are associated with increases in growth irradiance, and increases in temperature (decreases in nutrients) [11,12]. Additionally, the variability in our data generally confirms that $\theta$ is high under high light and nutrient-depleted conditions and low during low light and nutrient-repleted conditions in natural phytoplankton communities. In the northern part of SCS (Region I), the northeast wind was prevailing in the winter, which is characterized by cool, dry northeasterly flow. The momentum, heat, and buoyancy fluxes resulting from these atmospheric conditions promoted oceanic convective mixing that leaded to a cooler and saltier surface layer as the thermocline eroded and the mixed layer deepened over the winter [81,82]. This substantial nutrient pool was readily entrained

into the surface layer during the winter by the convective mixing and the value of $\theta$ was also decreased. Both increase of light and decrease of wind speed could increase the SST and enhance upper-ocean stratification in the summer. The upper layers of the ocean were rapidly depleted of nutrients with relatively high $\theta$. Therefore, we found that the linear relationship between $\theta$ and SST was significant in the northern SCS, as indicated by the relatively high values of $r$ being 0.53 ($p < 0.001$). Additionally, it was likely that $I_g$ was not the dominant factor causing $\theta$ change in the northern SCS, with no relationship between them ($r = 0.08$, $p = 0.81$). In Region II (central basin), the $\theta$ gradually decreased from spring to winter because of weakened light and reductions in nutrient stress (SST decrease); relatively high correlations between $\theta$ and $I_g$ and SST were detected, with the $r$ being 0.52/0.31 ($p = 0.09$/ $p < 0.001$), respectively (Figures 9b and 10b). However, there were no obvious correlations in the western part of SCS (Region III), as indicated by the low values of $r$ ($p > 0.1$), which could be due to the complexity of other processes, such as strong upwelling caused by the southwest monsoon during summer and eddy activity [75].

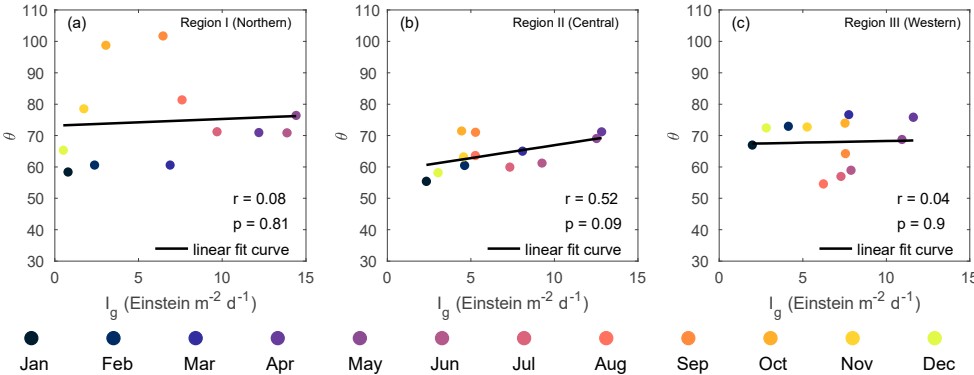

**Figure 9.** Monthly mean climatology of $\theta$ as a function of $I_g$. (**a**) Region I, (**b**) Region II, (**c**) Region III. The solid line represents a linear fit to OC-CCI data.

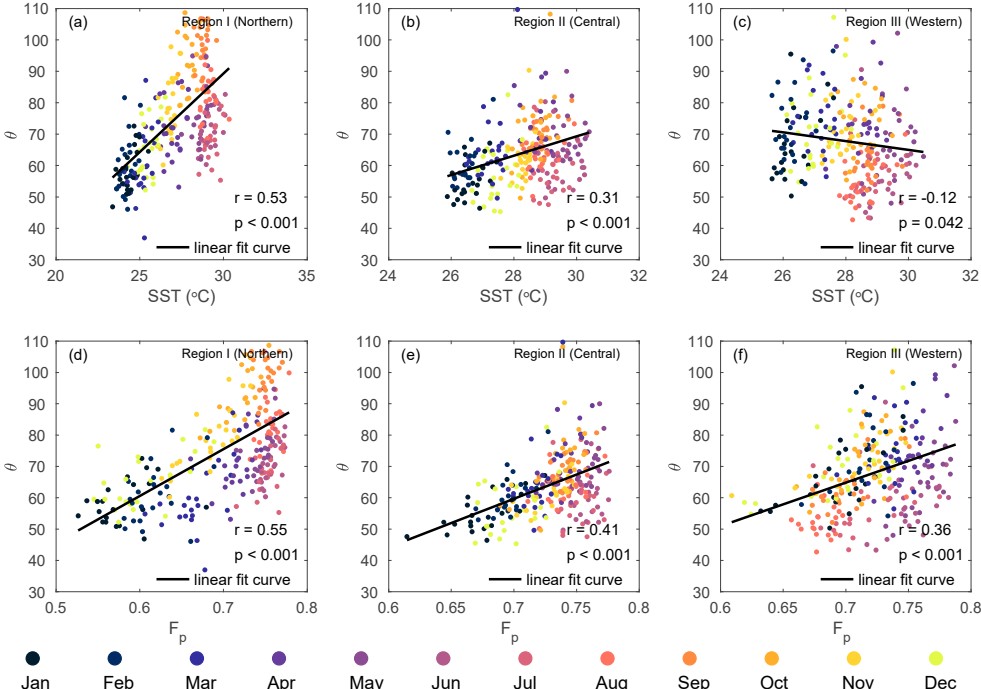

**Figure 10.** Monthly mean $\theta$ as a function of sea surface temperature (SST) and $F_p$. (**a,d**) Region I, (**b,e**) Region II, (**c,f**) Region III. The solid line represents a linear fit to OC-CCI data

The $\theta$ may also be affected by the phytoplankton community structure, being lowest for the larger diatom cells and highest for smaller species such as Prochlorococcus [30]. The fraction of picophytoplankton ($F_p$) was inferred from a three-component model that reparametrized many years of pigment measurements collected in the SCS [53], and their relationship with $\theta$ was studied (Figure 10d–f). As the fraction of $F_p$ increased, $\theta$ gradually increased except for the western part of SCS (Region III), as indicated by the relatively high value of $r$ being 0.64 ($p = 0.03$) (Region I), 0.58 ($p = 0.05$) (Region II), and by low value of $r$ being 0.24 ($p = 0.45$) (Region III). This relationship was consistent with current research results.

Additionally, we also found that the relationship between $\theta$ and environmental factors shows slight differences in different months, which may be due to a strong coupling relationship within environmental variables. From the scatter plot of $\theta$ versus environmental factors for different bioregions, we could only make simple speculations. The specific influencing process between them still needs further analysis.

## 5. Conclusions

In this study, phytoplankton carbon concentrations retrieved with an adjusted algorithm from ocean color satellite, were used as an alternative proxy for phytoplankton biomass in the South China Sea (SCS). The estimated phytoplankton carbon was comparable with results from the two previously established empirical models (Maranon et al. (2014) and Loisel et al. (2018)) in the literature. Additionally, the spatial and temporal variability of phytoplankton carbon and the carbon-to-chlorophyll a (Chl-a) ratio ($\theta$) were investigated by dividing the whole SCS into three subregions based on k-means algorithm. Drivers of their variability were further discussed by examining the effect of light intensity, temperature, and phytoplankton size structures.

Both phytoplankton carbon concentrations and $\theta$ showed significant seasonality in the SCS. Phytoplankton carbon concentrations were high in the northern and western SCS, but very low in the deep basin on the Philippine side of the SCS from November to February. From June to September, phytoplankton carbon concentrations were low throughout the SCS, with the exception of higher concentrations to the east of southern Vietnam. Over a 22-year period, the $\theta$ of individual pixels averaged 67.77 ± 32.44. The highest values were observed in September in the northern and central basin of SCS and in March in the western SCS. The lowest values were observed in January in the northern and central basin of SCS but in August in the west. It was found that spatial distribution of Chl-a concentrations was broadly similar to that of phytoplankton carbon, but some important differences remained. These differences can be indicated by the $\theta$, which varied over a wide range from <20 in eutrophic waters to >90 in oligotrophic regions in the SCS.

The robust verification for the use of satellite-derived phytoplankton carbon concentrations should include the collection of comprehensive in situ measurements in future work. Accurately resolving the significant spatiotemporal variability of $\theta$ in the SCS requires more field data related to phytoplankton physiological processes.

**Author Contributions:** Conceptualization, W.X., and G.W.; Formal analysis, W.X. and G.W.; Investigation, W.C., W.Z. and G.W.; Supervision, G.W., L.J. and X.C.; Writing—original draft, W.X.; Writing—review and editing, W.X., G.W., and L.J. All authors have read and agreed to the published version of the manuscript.

**Funding:** This research was funded by the National Natural Science Foundation of China (41776045, 41976170, 41976172), the Fundamental Research Funds for the Central Universities at Hohai University (2017B06714, B200201013), and the Natural Science Foundation of Jiangsu Province (Grants No BK 20200517).

**Data Availability Statement:** The data presented in this study are available on request from the corresponding author upon reasonable request.

**Acknowledgments:** We thank ESA OC-CCI for providing publicly available ocean colour data (https://esa-oceancolour-cci.org). We would like to thank the three referees for very constructive suggestions to improve the manuscript.

**Conflicts of Interest:** The authors declare no conflict of interest.

## Appendix A

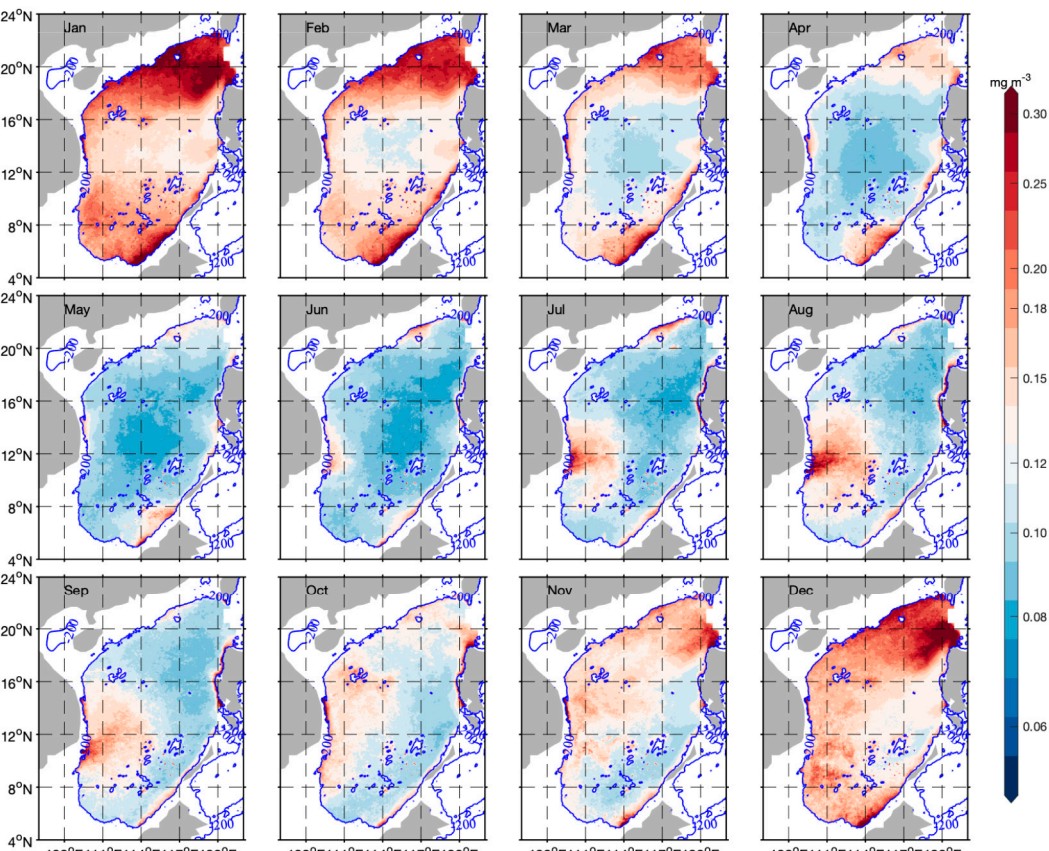

**Figure A1.** Monthly mean climatological distribution of the chlorophyll a (Chl-a) in the South China Sea (September 1997 to December 2019). The blue line shows the 200-m isobath.

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
