# Peer review of "Spatiotemporal Variability of Surface Phytoplankton Carbon and Carbon-to-Chlorophyll a Ratio in the South China Sea Based on Satellite Data"

_remotesensing, doi:10.3390/rs13010030_

Round 1

Reviewer 1 Report

This is the review of the work “Spatiotemporal variability of surface phytoplankton carbon and carbon-to-chlorophyll a ratio in the South China Sea based on satellite data” by Wenlong Xu and co-authors.

After having read the work I can recommend it for publication in Remote Sensing, after revision. I have made a series of comments, all meant to improve the quality of the work. In general the paper is well written although it is not always easy to understand what is part of the analysis and what is taken from the literature.

The introduction is well written and the line of reasoning clear. The objective of the work is set out explicitly, however, it could be refined (narrowed) based on the work major findings. Studying the space-time variability of phytoplankton biomass in the SCS is indeed a good (starting) objective; my suggestion, meant to improve the paper readability, is to narrow the objective towards the work major findings.

Materials and Method – this section needs more details; it is often not entirely clear what the authors have done precisely (see my comments below about the phytoplankton carbon retrieval), casting doubts on the overall significance of the results.

Introduction

Lines 52-57: does chlorophyll “unambiguously” represent phytoplankton biomass or is their relationship not straightforward?

Line 64: please replace “are covaried” with “are shown to covary” or “covary” or similar.

Line 65: please replace “estimated” with “estimate”

Line 85: please replace “have ever gave a conservative estimate” with “estimated” or “provided a conservative estimate”

Line 98: replace semicolon with comma

Materials and Methods

Figure 1: the figure should include a legend in which the various symbols used in the figure are explained. This information can be also included in the figure caption. This figure could be merged with Figure 2 or perhaps include the bathymetry, not just the 200m isobath.

Line 135: OC-CCI dataset is currently at version 5 and the authors may want to use state-of-the-art data and algorithms for their analysis.

Lines 132-145: it would be very useful if the authors would anticipate their use of kd and PAR, before equation 1. Moreover, it seems that PAR was derived from the algorithm used by GlobColour applied to OC-CCI reflectance, is my understanding correct? If so, why not using directly GlobColour PAR product? What is total absorption used for in this analysis?

Section 2.3: Since it is strongly acknowledged by the authors that the variability in phytoplankton chlorophyll-a represents both biomass variability and physiological adjustment to environmental factors, I wonder what would be the shape and temporal evolution of the clusters if bbp (as surrogate of carbon biomass) is used instead of chl-a.

Results

Section 3.1: it would be interesting to see, apart from the correlation value, some other statistics generally used in this kind of validation exercises such as root mean square difference, bias and relative errors. These numbers are also mentioned by the authors in relation to the validation of the QAA-derived bbp performed by Deng et al (2020, Optics Express). From these numbers, the authors would perhaps more clearly see what is already evident from the scatterplot, that the CCI-derived Chl-a (slightly or strongly?) overestimates in situ Chl-a. It would be also interesting to see the distribution of the matchups with respect to the k-means clusters and if the statistics do change significantly from one cluster to the other or if the performance of the CCI-derived Chl-a is stable across the entire domain.

Line 221: non-algal

Section 3.2

Shouldn’t this section or part of it belong to Materials and Method?

The authors state that their method follows Behrenfeld et al (2005), however there appear to be some important difference between the two approaches that the authors should set out more explicitly. Behrenfeld et al (2005) used bbp at 440 nm not at 555 nm. This discrepancy prevents the fit parameters (slope and intercept) to be straightforwardly compared (line 224) and bring to misleading conclusions (“As a result…” at line 225). However, from the caption of Figure 4, I can read that the authors did consider a spectral slope to adapt Behrenfeld et al (2005) curve to the bbp(555) axis. Is this spectral slope also applied in the comparison of the two bbpNAPs?

Moreover to build their figure 2, Behrenfeld et al (2005) used the regional monthly mean values (the monthly averages of the cluster areas), not the single pixel values, which is rather the approach followed by Bellacicco et al (2016 and 2018).

Figure 4: what is “our data”? it cannot be the in situ data (N=109), so it must be the CCI data; if so, please replace with OC-CCI data.

Lines 235-238: this is not a validation exercise, it is rather an evaluation of the consistency of the results, a comparison with what available from the literature.

Line 241: “slightly“ should rather be replaced with “significantly”

Figure 5: it would be better if the colors used to plot the 6 relationships were different, and so do the line styles

Lines 243-245: the “comparison” of the various algorithms shows that i) there is a certain degree of convergence with the algorithms of Marañon et al (2014) and Loisel et al (2018), ii) whereas the comparison with the two Sathyendranath et al (2009) and with Buck et al (1996) evidences significant differences. In both i) and ii), similarities and differences should be looked for in the methods or in situ data, representative of the biogeochemistry or trophic regime of the various areas, used to derive the various algorithms, matter of this qualitative comparison.

Section 3.3

Figure 8 – since the panels are used to compare the various regions, having the same axes would help also to highlight the low seasonality mentioned at lines 261-263.

Lines 272-274: this sentence must be supported by at least one reference or at least getting more into the rationale of this concept. I can understand that the Carbon contribution (biomass) of micro- is larger than that of pico-phytoplankton, and that chlorophyll concentration in micro-phytoplankton could be larger than that of pico-phytoplankton but still not as large as the Carbon difference between the two size fractions. If this is the rationale behind this concept it should still supported or better explained, especially because, as far as I have understood, most of the discussion in section 4.1 points in the opposite direction.

Line 278: “Those regions are defined in Figure 2a” – there is no need of this sentence, here.

Line 288: “The climatology monthly mean ? for each bioregion is displayed in Figure 8 b,d,f” – figure 8b,d,f has already been cited a few lines above, there is no need to introduce it here.

Line 290: “consistent” with what?

Discussion

Line 311: please mention that Xiu and Chai (2012) used both model and satellite data, the latter derived using the approach by Behrenfeld et al. (2005). The consistency found by the authors is not surprising and using it as a proof of truth without any other comment or caveat could be misleading.

Lines 319-321: this sentence does not take account of the plasticity of phytoplankton to short term variability of environmental factors.

Lines 317-326: I am getting confused, shouldn’t this be the other way around? Upwelling increases micro-phytoplankton capability to over compete pico-phytoplankton, which are characterized by higher ? (lines 272-274).

Line 335: “have been explained” should be replaced with “will be here explained”

Line 342: “climatology monthly mean” should be replaced by “monthly mean climatology”. There are several places in the manuscript in which this correction should be applied, please go through it.

Lines 343-345: increases in temperature and decrease in nutrient are indeed the same cause determined via SST analysis. The way it is presented looks like as if there were three independent parameters being compared with ?. Put “decrease in nutrients” in parentheses. A few comments:

  • The relationships in Figure 9 are null (region I and III) or weak (see r values) and their significance should be tested (T-Student?) and mentioned.
  • The mentioned relationships do not seem to happen at the same time (the order of monthly dots changes from a panel to the other), so that it is difficult to associate a unique set of conditions with space and time, e.g. with real conditions.
  • What it is rationale for using the anomalies rather than the climatological values?
  • Isn’t the three-component model by Lin et al (2014) matter of Materials and Methods? Surely more details about the model and its implementation are needed.
  • Figure 9c,f,i show that the C:Chl increases when small cells dominate, which further reinforces my confusion (see my comment about lines 317-326 and 272-274).
  • See my comment about “our data”
  • What is currently missing from the analysis is the explanation in terms of the process that determine what you showed from the data.

Line 348: “large values of r” when referring to r=0.65? r=0.65 means r2=0.4, not really this large.

Line 350: authors, when saying the correlation is significant, should mention at least which significance test they applied.

Line 354-357: the sentence do not add any significant value to the discussion. What the reader expect to see at this stage is an explanation of the process that drives the observed variability, at least conceptually. Then, that the exponential relationship found by Behrenfeld et al (2005) does not fit the data used here, is really of little relevance.

Conclusions

Lines 373-374: the results from the literature showed quite large variability and the relationship presented here is at one end of the range of variability.

Line 375: remove “(Chl-a)” and “(?)”

Lines 371-378: this is a summary of the methods

Lines 379-389: this is a summary of the results

Reviewer 2 Report

Certainly, the authors' statement on the importance of the problem of the ratio of the amount of carbon to the amount of chlorophyll in phytoplankton is correct. Anyway, the analyzes carried out by the authors on the example of a selected sea region confirm this. The results shown are not universal, as they concern a selected region, but they enrich the knowledge of spatial and seasonal changes in the carbon / chlorophyll index. They justify the need to develop this type of research.
In my opinion, the manuscript is worth publishing - the work is well organized and the conclusions are clear.
From my point of view, this manuscript does not require corrections (perhaps the editors will indicate the need to move web addresses from the text to the bibliographic list).

Reviewer 3 Report

The paper by Xu et al. discusses the results obtained by applying already available algorithms to satellite data in order to study the spatial and temporal variability of phytoplankton carbon and carbon-to-phytoplankton ratio in the South China Sea.

The paper is well written.

The motivation for using a regional algorithm for estimating the phytoplankton carbon concentration from remotely sensed data is satisfactory. The used approach is based on already existing algorithms applied to datasets available from ESA, NASA, NOAA, and others. 

The results are widely discussed, but most of the paper is just a description of values and trends from the obtained maps that have been divided into sub-regions using a k-mean clustering algorithm. Also, the majority of the abstract (from line 22 to 36) is focused on the results rather than presenting the data and the methods.

Some checks are necessary for the reference section since some references (i.e., ref. n. 1 and 13) contains words in upper case.
